# ADAPTIVE DATA AUGMENTATION WITH DEEP PARALLEL GENERATIVE MODELS

## ABSTRACT

Data augmentation(DA) is a useful technique to enlarge the size of the training set and prevent overfitting for different machine learning tasks when training data is scarce. However, current data augmentation techniques rely heavily on human design and domain knowledge, and existing automated approaches are yet to fully exploit the latent features in the training dataset. In this paper we propose an adaptive DA strategy based on generative models, where the training set adaptively enriches itself with sample images automatically constructed from deep generative models trained in parallel. We demonstrate by experiments that our data augmentation strategy, with little model-specific considerations, can be easily adapted to cross-domain deep learning/machine learning tasks such as image classification and image inpainting, while significantly improving model performance in both tasks.

## 1 INTRODUCTION

Deep learning and machine learning models produce highly successful results when given sufficient training data. However, when training data is scarce, overfitting will occur and the resulting model will be generalized poorly. It is also unavoidable that some data is very hard to retrieve. Data augmentation(DA) ameliorates such issues by enlarging the original data set and making more effective use of the information in existing data. Much prior work has centered on data augmentation strategies based on human design, including heuristic data augmentation strategies such as crop, mirror, rotation and distortion Krizhevsky et al. (2012); Simard et al. (2003), interpolating through labeled data points in feature spaces DeVries & Taylor (2017), and adversarial data augmentation strategies based on Teo et al. (2008); Fawzi et al. (2016). These methods have greatly aided many deep learning tasks across several domains such as classification Krizhevsky et al. (2012), image segmentation Yang et al. (2017) and image reconstruction/inpainting Alvarez-Gila et al. (2017).

Despite their success, these DA methods generally require domain-specific expert knowledge, manual operations and extensive amount of tuning depending on actual contexts Ciresan et al.; Dosovitskiy et al. (2016). In particular, the need to directly operate on existing data with domain knowledge prevents many previous data augmentation strategies from being applicable to more general settings. To circumvent the need for specific domain knowledge in data augmentation, more recent work Antoniou et al. (2017) utilizes generative adversarial networks(GANs) Goodfellow et al. (2014) to produce images that better encode features in the latent space of training data. By alternatively optimizing the generator $G$ and the discriminator $D$ in the GAN, the GAN is able to produce images similar to the original data and effectively complement the training set. It has been shown in experiments Antoniou et al. (2017) that GAN-based methods have indeed significantly boosted the performance of classifiers under limited data through automatic augmentation, but applications into other tasks are yet to be explored. Furthermore, given the computational complexity and difficulty in GAN training, a natural way to promote scalability is to consider parallelism Intrator et al. (2018); Durugkar et al. (2016).

In view of these considerations, we propose in this paper a new adaptive DA strategy based on deep generative models. Under such a framework, the original training set adaptively enriches itself with sample images automatically constructed from Generative Adversarial Networks (GANs) trained in parallel. Our contributions in this paper can be summarized as follows:

- We propose a general adaptive black-box data augmentation strategy to diversify enhance training data, with no task-specific considerations.

- We also include in our model a novel $K$-fold parallel framework, which helps make the most use of the existing data.

- Experiments over various datasets and tasks demonstrate the effectiveness of our method in different contexts and tasks.

## 2 RELATED WORK

Augmenting data from limited amounts of training data has been a persistent challenge within the machine learning community. Here we summarize the related work in different categories.

**Data Augmentation(DA)** Previous work on data augmentation can be classified into several groups. Traditional Heuristic DA strategies, such as crop, mirror, rotation and distortion Krizhevsky et al. (2012); Simard et al. (2003), have found their way in many deep classification tasks, but these methods generally require domain-specific expert knowledge, manual operations and extensive amount of tuning depending on actual contexts Ciresan et al.; Dosovitskiy et al. (2016). Alternative DA methods consider interpolation through labeled data points in feature spaces DeVries & Taylor (2017), but their dependence on class labels makes them inapplicable for tasks with weak or no supervision. Adversarial Data Augmentation strategies Teo et al. (2008); Fawzi et al. (2016) choose from a select number of transformation operations to maximize the loss function of the end classification model involved in the task. While good motivations for our methods, these methods make strong assumptions over the types of augmentation and are difficult to generalize. Ratner et al. (2017); Cubuk et al. (2018) transform the problem of choosing data augmentation strategies into a reinforcement learning policy search problems, but the choice of augmentation methods are still limited and the reinforcement learning algorithms have non-trivial computation overhead in addition to the main task.

**ML problems with limited data** For classfication with limited samples, Salamon & Bello (2017) proposed a convolutional neural network(CNN) to classify environmental sounds with limited samples. Other algorithms have been proposed in Frid-Adar et al. (2018); Zhu et al. (2017), yet many of them have assumptions/constraints that hurts their capacity for generalization. For unsupervised learning models, recent research on sample complexity reduction in GAN training seeks to reparametrize the input noise using variational inference Gurumurthy et al. (2017); Nowozin et al. (2016), but this method has severe mathematical limitation that prevents further generalization. Wang et al. (2018) adopts transfer learning techniques to train a new GAN for limited data from a pre-trained GAN network. While effective, this approach requires a pre-trained network in the first place and doesn't apply to the cases when data is scarce.

**Adaptive and Distributed Learning** In recent years, adaptive learning has achieved good results in many training tasks. Yu et al. (2018), Clavera et al. (2018) have proposed adaptive learning in reinforcement learning and meta-learning, and the strategies described within motivate our methods in this paper. On the side of deep learning, distributed multi-discriminator models Intrator et al. (2018); Durugkar et al. (2016) also enhance the performances of generative algorithms,yet these models require large datasets to train and have great computational complexity. Moreover, these models are trained on fixed given datasets, meaning they are still susceptible to the inherent biases of the training data.

**Deep learning with Limited Data** Salamon & Bello (2017) proposed an algorithm to classify environmental sounds with limited samples, through implementing a Convolutional Neural Network with augmented samples. Training GANs on limited amounts of training data has been a persistent challenge within deep learning community. Previous efforts to address the issue of labeled data scarcity generally fall into two groups: optimizing the structures of GANs to allow for better feature representation of data, and augmenting the training data through techniques.

Along the first line of research, prior research optimized the GAN in Radford et al. (2015) by considering stronger mathematical objectives for more powerful latent space representation in general Nowozin et al. (2016); Arjovsky et al. (2017); Gulrajani et al. (2017); Mroueh & Sercu (2017). In addition, recent research on GANs Gurumurthy et al. (2017); Nowozin et al. (2016) reparametrized the input noise using variational inference Kingma & Welling (2014) by assuming that the latent

space could be modeled by a tractable prior, but noise reparametrization has severe mathematical limitation that prevents applicability to more general models.

# 3 PAGDA: PARALLEL ADAPTIVE GENERATIVE DATA AUGMENTATION

In this section we describe the details of Parallel Adaptive Generative Data Augmentation(PAGDA). Our method consists of three interrelated components: generative data augmentation, parallel image generation with fold division, and adaptive weight adjustment.

## 3.1 GENERATIVE DATA AUGMENTATION

To ensure that make full use of the information contained in the existing images, the first part of our method involves *generative data augmentation*, which constructs varied images given the training set by repeatedly generating samples from and adding samples to the training set using a generative adversarial net.

We start off with a limited training set, and consecutively run the generative adversarial net using the set. After running a fixed number $t$ of regular training epochs, we proceed to the augmentation epoch where the augmentation is conducted. During the augmentation epoch, we extract a number of sample images from the generator $G$ using standard procedures of sample image generation as described in Radford et al. (2015); Gulrajani et al. (2017). For this batch of samples, we calculate the Inception Score(IS) as defined by Salimans et al. (2016) to measure the authenticity of the images generated, which we denote as $w$ and is described as follows:

$$w = \exp(\mathbb{E}_{\mathbf{x}}(D_{\mathrm{KL}}(p(y|\mathbf{x})\|p(y)))),$$

Here the Inception Score (IS) measures the entropy of generated images, with higher scores indicating greater diversity. It thus provides a metric that evaluates the power of generator to produce realistic images $y$ from the distribution of original real images $\mathbf{x}$: the higher the value of $w$, the more power the corresponding generator $G$. This batch of images are then added back into the original training set for subsequent augmentation epochs. We alternate running $t$ regular training epochs and the augmentation epoch for a fixed number of times or until convergence. Figure 1 is a flow-chart of our procedure.

Notice that our procedure is agnostic to the specific architecture of generative adversarial net used to augment the training data. Since GANs capture the information in the latent feature space of the images and translate such information into generated images, our method has the capacity to reveal the potential features that are possibly not visually evident in the original training images. Moreover, compared with many other data augmentation strategies which require one to pre-define the operations to be carried on the images, our method automatically enriches the training set and does not require human intervention.

## 3.2 PARALLEL FOLD DIVISION

The second part of our method consists of a parallel data generation strategy, inspired by $K$-fold cross validation in machine learning Bishop (2006). Dividing the training data into $K$ folds at the beginning, we run in parallel $K$ independent generators $\{G_i\}_{i=1}^{K}$. Each generator $G_i$ is trained on one of data groups, and each data group $i$ consists of $K-1$ folds of the training set, except for the $i$-th fold. After images are generated in each generator $G_i$ in the training epochs, the sample images produced by each generator during the augmentation epoch are fed back into the respective training data groups. To allow for maximal usage of each generated image, we insert the images in a way such that the images generated by one generator $G_i$ are sent to the training data groups corresponding to all other $K-1$ generators except for that corresponding to $G_i$. This is to insure that the different generators in parallel have access to as many varied data pieces as possible in subsequent steps of training, so as to prevent overfitting and bolster the robustness of our strategy. Figure 2 demonstrates our algorithm.

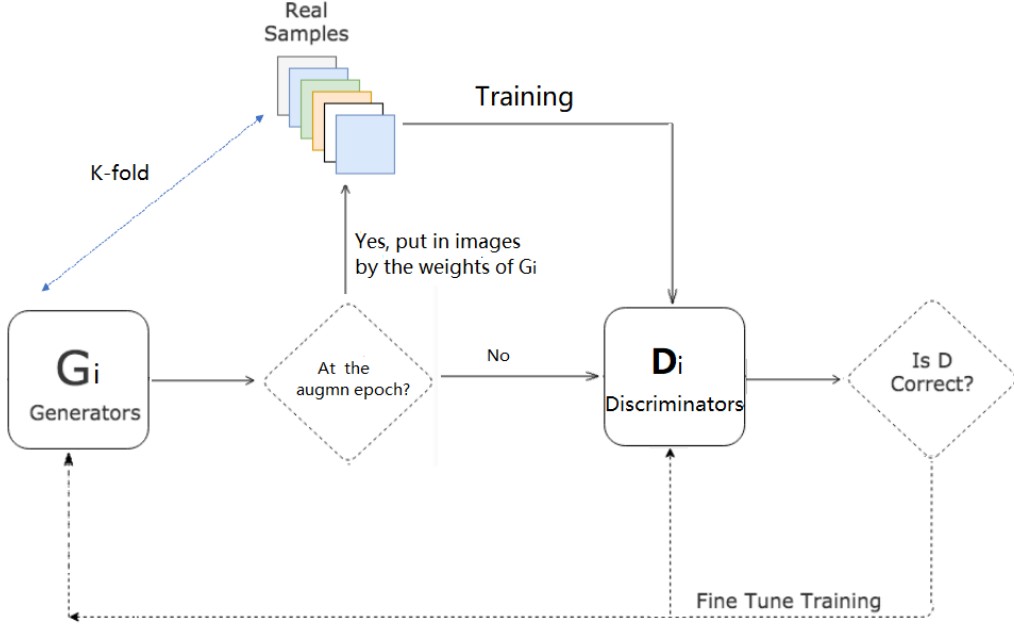

Figure 1: Generative Data Augmentation

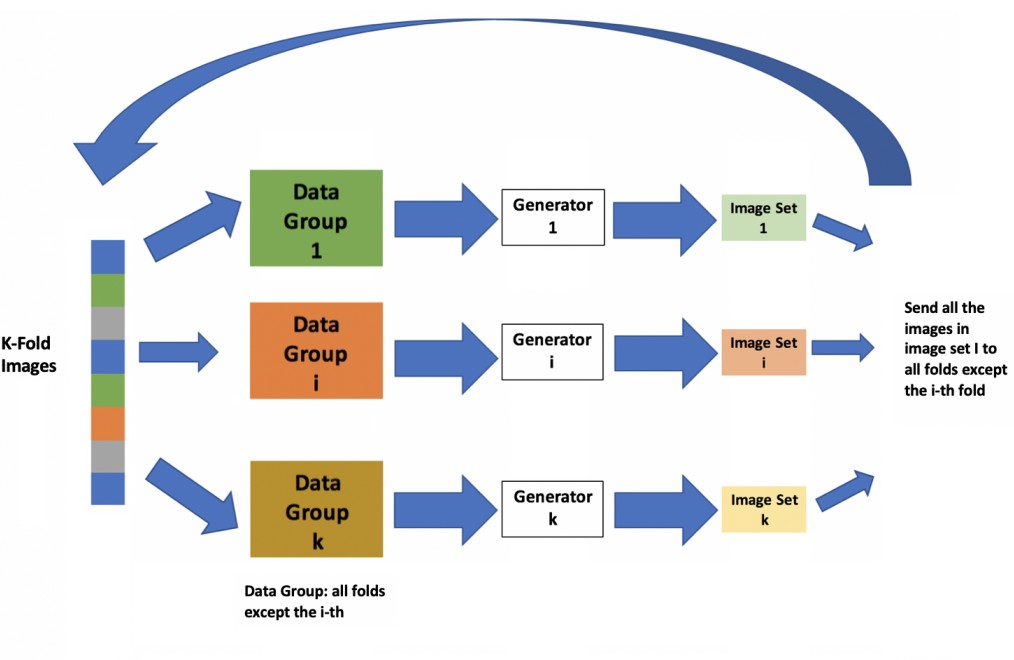

Figure 2: Parallel Image Generation

## 3.3 ADAPTIVE GENERATOR WEIGHTING

Furthermore, to determine which generators are the most effective in generating authentic images, we introduce *adaptive generator weighting* at each augmentation epoch. At the initial stage, all the generators are treated equally. Before the batch of sample images generated by one generator $G_i$

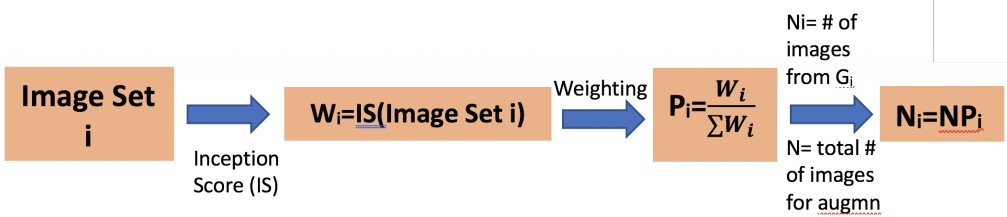

Figure 3: Adaptive Generator Weighting

are sent to the data group corresponding to other $K - 1$ generators, we collect the inception scores $\{w_i\}_{i=1}^K$ computed in section 3.1. Since higher inception scores imply better performance of the generator, we define *the generator weight* $p_i$ of a generator $G_i$ as

$$p_i = \frac{\exp(w_i)}{\sum_{j=1}^K \exp(w_j)},$$

and use this weight to determine how many images should be sampled from generator $G_i$ to be sent to other data groups for subsequent training in the very next augmentation epoch. When the total number of samples to be collected from generators are fixed, this method enables generators with better realistic image generation power to contribute more to the future training data groups. More realistic training sets thus augmented, in turn, exert more positive influence on the images to be generated. Algorithm 1 illustrates the whole 3-step process, which we denote as PAGDA(Parallel Adaptive Generative Data Augmentation).

---

**Algorithm 1** Parallel Adaptive Generative Data Augmentation

**Input:** A limited dataset $\mathcal{D} = \{(x_i, y_i)\}_{i=1}^N$
**Output:** An augmented dataset $\mathcal{D}' = \{(x_i, y_i)\}_{i=1}^{N'}$
1: **procedure** PAGDA($\mathcal{D}$)
2:     Divide $\mathcal{D}$ into $K$ equal size folds $\{\mathcal{D}_i\}_1^K$;          ▷ Parallel Fold Division
3:     Initialize a Generative Adversarial Network $G_i$ on each $\mathcal{D}_i$, and all GAN weights $p_i = 1$;
4:     Fix size of augmentation batch at $n_0$;
5:     **while** Training **do**
6:         Train $G_i$ on $\mathcal{D}_i$ for a certain number $t$ of epochs;
7:         Extract $S_i = n_0 p_i$ images from discriminator of $G_i$;    ▷ Generative Data Augmentation
8:         $w_i$ = Inception Score of Imageset(i);
9:         $p_i = \frac{\exp(w_i)}{\sum_{j=1}^K \exp(w_j)}$;          ▷ Adaptive Generator Weighting
10:       $D_i = D_i \cup (\cup_{k \neq i} S_k)$;
11:     **return** $\mathcal{D}' = \mathcal{D} \cup (\cup_{i=1}^K D_i)$

---

Note that all three strategies introduced go hand in hand, with no need for model specific considerations. As demonstrated by our experiments Section 4, training different GANs in parallel from different folds of data substantially boosts the quality of the training set and that of the generated images.

## 4 EXPERIMENTS

### 4.1 MULTITASK EXPERIMENTAL SETTINGS

To illustrate the effectiveness of our adaptive model, we applied our model for multiple machine learning tasks, we have applied our data augmentation method to two tasks: image classification and image inpainting.

For image classification we constructed our dataset from Imagenet and Cifar-10 by randomly drawing 5000 images from each dataset respectively and applied adaptive generative models on these

reduced datasets. The augmented datasets are then used to train an AlexNet CNN classifier, and the classification results are compared with the results obtained from an AlexNet trained on the corresponding original unaugmented datasets or augmented datasets with different strategies.

For image inpainting, we constructed our reduced sub-datasets from open dataset named Places. The whole Places dataset consists of different sub datasets for images with different scenarios (e.g. ocean sub datsets contains images with all beautiful scenes from the sea). We randomly pickup images of 3 scenarios (ocean, orchard, piers) to obtain a reduced sub dataset. Each sub dataset contains 5000 images.

To ensure the parallelism of the experiments, we trained our model in a multi-gpu environment to make simultaneously training. Under such a setting, all the data groups are trained at the same time, and each GAN model corresponding to each data group is trained in a separate thread. All of our experiments are conducted on 2 servers with 8 Tesla-V GPU (32GB RAM, 7.8 TeraFLOPS) and Intel® Xeon® Processor E5 (2.00 GHz).

## 4.2 EVALUATION ON CLASSIFICATION FROM IMAGE DATASETS

For the task of image classifidation, we have conducted extensive experiments on different image datasets to illustrate the effect of our data augmentation strategy. For our experiments on classification, we first augment the reduced Cifar-10, reduced Imagenet datasets, and then train the CNN classifier with the augmented dataset.The classifier accuracies with and without augmentation are listed in Table 1 below with the experimental setting described in the next paragraph.

Table 1: Classification Accuracy with different augment strategy

| Datasets | w/o augment | image flip | image crop | with augment |
|---|---|---|---|---|
| Reduced Cifar-10 | 74.7 | 75.3 | 75.6 | **81.3** |
| Reduced Imagenet | 78.7 | 78.9 | 78.8 | **88.1** |

In our experiments, we augment the training set with 1000 noised images every 100 training epoches, and repeat the procedure 5 times. By comparison, the unaugmented GAN is run over the same initial training data, with the number of epochs the same as the product of 100 and augmentation times. For optimal choice of fold number $K$, we varied $K$ from 2 to 10, examined the level of accuracy using different values of $K$, and choose 5 to produce the yield the best classification results as listed in table 1. Our observation is largely consistent with the heuristics of K-fold cross validation in machine learning, where $K$-values between 4-5 are often the optimal Bishop (2006).

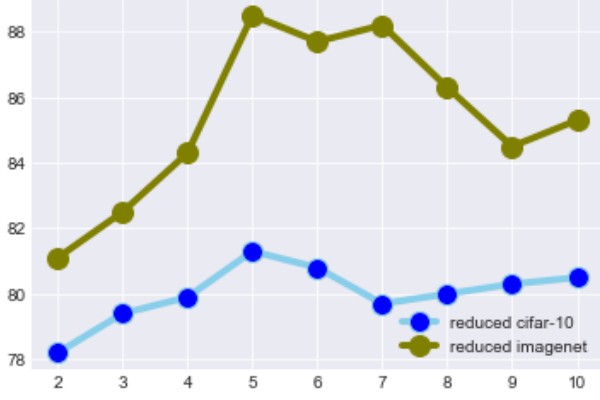

Figure 4: Accuracy with different number of folders

## 4.3 EVALUATION ON NEW IMAGE GENERATION

Along with the task of classification, we considered the task of new image generation from given training images. We have applied similar settings as in image classification, and the goal is to

produce images with higher resolution and more authentic semantics with generative adversarial networks. In our setting, we first train the state-of-the-art WGAN to produce images from an unaugmented dataset, and repeat the procedure on an image data set that has been augmented.

Figures 5,6,7,8,9 are some sample images that our method has produced compared with those produced without data augmentation. We observe that GANs with parallel recurrent image augmentation produce semantically coherent and visually diverse images earlier than the unaugmented GANs, while able to avoid fluctuations seen in unaugmented GANs during training.

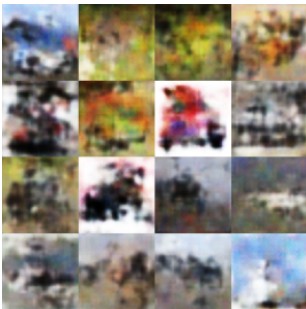 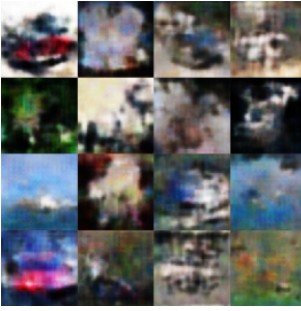 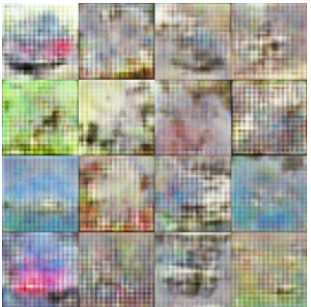

Figure 5: Augmented GAN at Epoch 271,Reduced-CIFAR    Figure 6: Unaugmented GAN at Epoch 361,Reduced-CIFAR    Figure 7: Unaugmented GAN at Epoch 342, Reduced-CIFAR

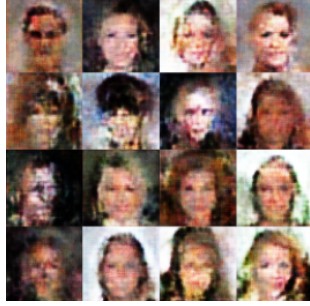 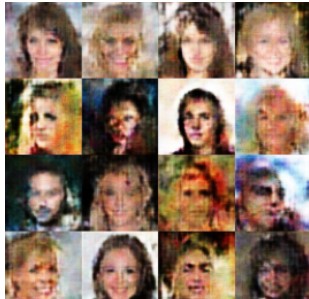

Figure 8: Unaugmented GAN, CelebA    Figure 9: Augmented GAN, CelebA

### 4.4 EVALUATION ON INPAINTING

For the task of inpainting, we augment the reduced Places dataset constructed in the experiment. As we mentioned 4.1, we apply 3 subset named with Ocean, Orchard and Piers, each subset contains 5000 images. Without loss of generality, we train a WGAN-GP model as generative model for inpainting from the augmented dataset. We then select testing images that are not selected in the training set, and add to them gray masks covering the center part of these images. We then applied our trained WGAN-GP to generate patches that cover the masked portion of the inpainting image.

For qualitative analysis, figure 10 lists a couple of generated images with and without augmentation. The semantic coherence within visual comparisons demonstrate the effectiveness of our method.

For a quantitative evaluation of quality for the images generated by our augmentation method as compared with the conventional methods such as image flipping and image cropping, we use the Inception Score(IS) Salimans et al. (2016) described previously as well as Frechet Inception Distance(FID) Heusel et al. (2017) for metrics. IS measures the entropy of generated images, with higher scores indicating greater diversity. On the other hand, FID measures the distance between the generated data and real data with two respective means and variances. Thus, the larger the IS and the smaller the FID, the better the performances of the model. An alternative measure involving the exponent of inception score has been proposed in Gurumurthy et al. (2017); given the diversity and semantic complexity of images involved in our experiments, we will stick to the original formulation as proposed in Salimans et al. (2016).

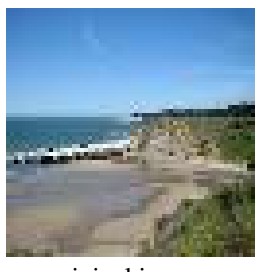 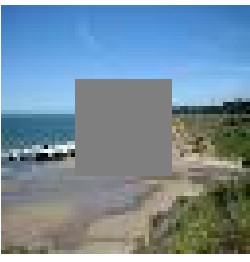 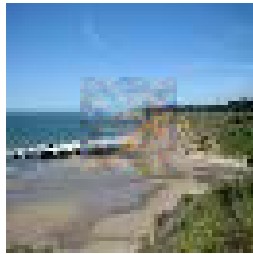 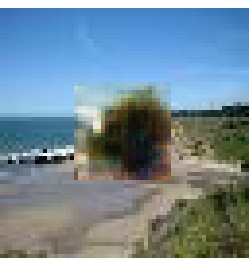

original image            masked image            with PAGANDA            w/o PAGANDA

Figure 10: Inpainting results on reduced datasets

The inception score(IS) and Frechet Inception Distance (FID) we used in all experiments are calculated by Python scripts available respectively at `https://github.com/openai/improved-gan/tree/master/inception_score` and `https://github.com/bioinf-jku/TTUR`.

Table 2: IS/FID scores of on Places with different Data Augmentation strategies

| Subsets | Noise | Image Flip | Image Crop | Our Strategy |
|---|---|---|---|---|
| Ocean | 4.31/102.7 | 4.33/107.8 | 3.99/114.2 | **5.75/79.3** |
| Orchard | 4.39/105.3 | 4.28/106.4 | 3.97/113.2 | **5.83/82.4** |
| Pier | 4.34/107.1 | 4.30/109.8 | 4.00/109.3 | **5.79/90.2** |

Table 2 lists the results of data augmentation based on different subsets of Places dataset. We have conducted the set of experiments with varying numbers of parallel folders and have discovered that the value K=5 has helped achieve the best image quality. Notice that the images generated with our method are produced with the state-of-the-art WGAN-GP as each parallel generative model. Here in addition to considerations in K-fold Cross ValidationBishop (2006), our choice of $K$-values are motivated by distributed MD-GAN as described in Hardy et al. (2018), where $K$-values ranging from 3 to 10 are shown to achieve good performances. Table 2 list IS and FID of results experimented on different sub datasets with conventional data augmentation. Clearly, adaptive sample augmentation produces better images as measured quantitatively and some other simple augmentation strategies.

## 5 CONCLUSION AND FUTURE WORK

We have proposed in this paper a parallel adaptive based data augmentation strategy, and demonstrate through experiments and analysis that our strategy significantly improves the performances of existing models on a variety of tasks, outcompeting many traditional methods aiming towards similar goals. In addition, our paper shows that our adaptive data augmentation strategy effectively requires little task-specific considerations, since the proposed framework can easily be adapted to a variety of generative models and ML paradigms. Our strategy is not only simple to implement, but also demonstrates capability on multiple tasks in machine learning and computer vision, since it does not require specific information about the task being analyzed.

As a further step, we are investigating the relationship between our proposed approach and other established methods. One possible pathway, for instance, lies in the DA strategy based on reinforcement learning as described in Cubuk et al. (2018) that gives more control to image generation via reward designation. We believe such motivation would inspire better design of the adaptive generator weight functions in our paper. We also hope to apply our idea to other generative models such as VAE in Kingma & Welling (2014), and further optimize our strategy using recent theoretical advances, and wish to investigate the scenarios where the tasks involved are interrelated. Application wise, we are aiming to extend our parallel GAN model to medical image synthesis/generation, since the amount of data available for machine learning is often highly limited. Another goal is to extend the scope of application from CV/DL to more general data mining, where alternative evaluation metrics can be used for designing the generator weight function in our model.

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
