# OpenReview forum: "Adaptive Data Augmentation with Deep Parallel Generative Models"
_ICLR.cc/2020/Conference — Reject_

### Official Review · AnonReviewer2 · 2019-10-18
**Official Blind Review #2**

**Rating:** 1

**Review:**

The paper propose a heuristic for making data augmentation more "end to end" rather than an ad hoc technique. The idea is to train multiple GANs on subsets of the data, and once in a while use these to generate new data aid the training of the final model.

I found that the justification for the proposed heuristic was largely missing. Why is the proposed method good?Is it optimal in any sense? Does it correspond to a specific model assumption? Is there an intuition why it's a good idea?

Intuitively, I would expect that it is hard to train a GAN (or another generative model) than it is to train a classifier, so why is it s a good idea to augment the dataset using GANs?

One of the key tricks of the paper is to split the data in multiple folds and train one GAN per fold. I would expect this to be fairly unstable is building generative models tend to be very "data demanding" (Wuch that working with only 1/5 of the total data could be problematic).

In recent years there have been quite some work on learning of data augmentation which isn't cited in the paper. I'd recommend looking at

  "Dreaming More Data: Class-dependent Distributions over Diffeomorphisms for Learned Data Augmentation", Hauberg et al., AISTATS 2016.
  "A Bayesian Data Augmentation Approach for Learning Deep Models", Tran et al. NeurIPS 2017

and the references therein to get a better coverage of previous work.

I generally find it difficult to assess the presented experiments. Since the approach is deemed general, why isn't it applied to general tasks with an established ground truth? E.g. why isn't the technique applied to classification or regression? That would make it much easier to assess if the approach does something sensible. I am also missing elementary baselines, e.g. the usual hand-crafted data augmentation should also appear in the baselines.

**Experience Assessment:**

I have published one or two papers in this area.

**Review Assessment: Checking Correctness Of Derivations And Theory:**

N/A

**Review Assessment: Checking Correctness Of Experiments:**

I assessed the sensibility of the experiments.

**Review Assessment: Thoroughness In Paper Reading:**

I made a quick assessment of this paper.

---

### Official Review · AnonReviewer3 · 2019-10-23
**Official Blind Review #3**

**Rating:** 1

**Review:**

This paper proposes a new approach to data augmentation using GANs, called Parallel Adaptive Generative Data Augmentation (PAGDA), in which the following procedure is used:
- (1) First, GANs are used to generate different batches of data points based on the starting training set, and then these batches are added back to the training set
- (2) This is done using an ensemble of K GANs where each GAN is trained on K-1/K portion of the training set, and the resulting samples are sent to the training sets of the other K-1 GANs
- (3) The generators are sampled from proportional to their (exponentially normalized) inception scores

Overall, this paper unfortunately should be rejected due primarily to inadequate experimental evaluation.  Specifically, the paper proposes a new method (and a very complex and seemingly arbitrary one at that), in a space where there are many pre-existing approaches, and then neither (a) compares to any pre-existing baselines or approaches (other than the most trivial of manual data augmentation strategies), nor (b) experimentally justifies or explores any of the design choices in the extremely convoluted method design.  Therefore, there is no real way for the reader to render any conclusion about the merit of the proposed approach: how does it compare to many (seemingly similar) existing approaches?  Why were the various design decisions made, which ones are more or less important?  How well does it work?  No real way to answer these given the extremely minimal experimental section.

Additionally, there is relatively little detail in the main description of the method (e.g. how is the GAN trained on labeled data?  How exactly is the sampling done?).  Also, worth noting that "Multi-task" is used incorrectly in the experiments section.

**Experience Assessment:**

I have published one or two papers in this area.

**Review Assessment: Checking Correctness Of Derivations And Theory:**

N/A

**Review Assessment: Checking Correctness Of Experiments:**

I assessed the sensibility of the experiments.

**Review Assessment: Thoroughness In Paper Reading:**

I read the paper thoroughly.

---

### Official Review · AnonReviewer1 · 2019-10-24
**Official Blind Review #1**

**Rating:** 1

**Review:**

This paper proposes an method based on multiple GANs trained on different splits of the training data in order to generate additional samples to include to the training data. The number of added samples from each GAN depends on the quality of the generation based on their inception scores. This approach helps to improve the performance of a convolutional neural network on different tasks such as image classification, image generation and image inpainting.

I rated this paper as weak reject because this paper is weak on several aspects
- In related work many similar approaches are missing (see below).
- In the methodology the actual formulation for the GAN is not presented.
- The main novelty of the paper seems to be the use of multiple GAN on different splits of the data, which seems a bit limited.
- The experimental evaluation is limited (see below).

Related work:
Several works with very similar intentions have been overlooked:
("A Bayesian Data Augmentation Approach for Learning Deep Models, Toan Tran, Trung Pham, Gustavo Carneiro, Lyle Palmer, Ian Reid) uses GAN during training for generating samples for data augmentation.
("Dada:  Deep adversarial data augmentation for extremely low data regime classification", Xiaofeng Zhang,  Zhangyang Wang,  Dong Liu,  and Qing Ling) uses another GAN model for generating samples on low data regime for data augmentation
("Adversarial Learning of General Transformations for Data Augmentation", Saypraseuth Mounsaveng, David Vazquez, Ismail Ben Ayed, Marco Pedersoli), uses GAN for generating samples for data augmentation on reduced datasets.
(Triple Generative Adversarial Nets, Chongxuan Li, Kun Xu, Jun Zhu, Bo Zhang) uses a gan model for generating samples for semi-supervised learning with few labelled samples.
(Triangle Generative Adversarial NetworksZhe Gan∗, Liqun Chen∗, Weiyao Wang, Yunchen Pu, Yizhe Zhang,Hao Liu, Chunyuan Li, Lawrence Carin) in which again GAN is used to improve in semi-supervised settings.

Experimental evaluation:
In the experimental evaluation the importance of the use of k-fold is shown only on Fig.4. Additionally, in table 1 the method is compared with weak baselines (no DA, flip, crop, but not the combination of flip and crop which is standard on CIFAR10) and it is not compared with any other approach (and there are several as shown in related work).
For generating images, no quantitative values are provided, just some generation examples.

**Experience Assessment:**

I have published one or two papers in this area.

**Review Assessment: Checking Correctness Of Derivations And Theory:**

I assessed the sensibility of the derivations and theory.

**Review Assessment: Checking Correctness Of Experiments:**

I assessed the sensibility of the experiments.

**Review Assessment: Thoroughness In Paper Reading:**

I read the paper thoroughly.

---

### Decision · Program_Chairs · 2019-12-19

**Decision:**

Reject

**Comment:**

This paper proposes a data augmentation method based on Generative Adversarial Networks by training several GANs on subsets of the data which are then used to synthesise new training examples in proportion to their estimated quality as measured by the Inception Score. The reviewers have raised several critical issues with the work, including motivation (it can be harder to train a generative model than a discriminative one), novelty, complexity of the proposed method, and lack of comparison to existing methods. Perhaps the most important one is the inadequate empirical evaluation. The authors didn’t address any of the raised concerns in the rebuttal. I will hence recommend the rejection of this paper.